# Symptomatic Uterine Rupture: A Fifteen Year Review

**DOI:** 10.3390/medicina56110574

**Published:** 2020-10-29

**Authors:** Egle Savukyne, Raimonda Bykovaite-Stankeviciene, Egle Machtejeviene, Ruta Nadisauskiene, Regina Maciuleviciene

**Affiliations:** Department of Obstetrics and Gynaecology, Medical Academy, Lithuanian University of Health Sciences, Kaunas LT-50161, Lithuania; raimonda.bykovaite@stud.lsmu.lt (R.B.-S.); egle.machtejeviene@lsmuni.lt (E.M.); ruta.nadisauskiene@lsmu.lt (R.N.); regina.maciuleviciene@lsmu.lt (R.M.)

**Keywords:** uterine rupture, vaginal birth after previous cesarean section, induction of labour, augmentation of labour, risk factor

## Abstract

*Background and objectives:* To assess the incidence of complete and partial uterine rupture during childbirth in a single tertiary referral centre as well as the significant risk factors, symptoms and peripartum complications. *Materials and Methods:* A retrospective single-centre study involved all cases of uterine rupture at the Kaunas Perinatal Centre in 2004–2019. Data were from a local medical database complemented with written information from medical records. We included 45,893 women with an intact uterus and 5630 with uterine scars. Women (*n* = 5626) with scarred uterus’ after previous cesarean delivery. The diagnosis was defined by clinical symptoms, leading to an emergency cesarean delivery, when complete or partial uterine rupture (*n* = 35) was confirmed. Asymptomatic cases, when uterine rupture was found at elective cesarean section (*n* = 3), were excluded. The control group is represented by all births delivered in our department during the study period (*n* = 51,525). The outcome was complete (tearing of all uterine wall layers, including serosa and membranes) and partial uterine rupture (uterine muscle defect but intact serosa), common uterine rupture symptoms. Risk factors were parameters related to pregnancy and labour. *Results:* 51,525 deliveries occurred in Kaunas Perinatal Centre during the 15 years of the study period. A total number of 35 (0.06%) symptomatic uterine ruptures were recorded: 22 complete and 13 partial, leading to an incidence rate of 6.8 per 10,000 deliveries. The uterine rupture incidence rate after a single previous cesarean delivery is 44.4 per 10,000 births. 29 (83%) cases had a uterine scar after previous cesarean, 4 (11%) had a previous laparoscopic myomectomy, 2 (6%) had an unscarred uterus. The most significant risk factors of uterine rupture include uterine scarring and augmentation or epidural anaesthesia in patients with a uterine scar after cesarean delivery. The most common clinical sign was acute abdominal pain in labour 18 (51%). No maternal, six intrapartum perinatal deaths (17%) occurred, and one hysterectomy (2.8%) was performed due to uterine rupture. Neonatal mortality reached 22% among the complete ruptures. Average blood loss was 1415 mL, 4 (11%) patients required blood transfusion. *Conclusions:* The incidence rate of uterine rupture (complete and incomplete) at Kaunas Perinatal Centre is 6.8 per 10,000 deliveries. In cases with a scar of the uterus after a single cesarean, the incidence of uterine rupture is higher, exceeding 44 cases per 10,000 births. The most significant risk factors were uterine scar and augmentation or epidural anaesthesia in a previous cesarean delivery. Acute abdominal pain in labour is the most frequent symptom for uterine rupture.

## 1. Introduction

Uterine rupture is a rare but dangerous peripartum complication that occurs in 1 out of 280–12,000 births [1,2]. This can lead to fetal death, hysterectomy, or even maternal death due to massive bleeding. According to the literature, scarring of the uterus after previous cesarean section (CS) and induction of labour using prostaglandins are the most common uterine rupture risk factors [1,3,4]. The growing number of CSs worldwide encourages us to remember the problem of uterine ruptures more often [5]. CS rates increased from 9.6% in 1995 to 25% in 2011 in Lithuania. After the start of using the 10-group Robson classification as a national audit tool, the rate of CS decreased to 20.8% in 2017 [5]. Classically, uterine rupture signs are represented with fetal heart rate abnormalities, abdominal pain, and vaginal bleeding. The purpose of this study was to assess the frequency of uterine ruptures in a tertiary referral centre, to identify risk factors and symptoms for complete and partial uterine rupture in labour, common symptoms of uterine rupture and peripartum complications.

## 2. Materials and Methods

A retrospective study was conducted to analyze uterine ruptures that occurred at the Department of Obstetrics and Gynecology of the Hospital of Lithuanian University of Health Sciences (LUHS) *Kauno klinikos*. The department is a tertiary referral centre where mainly high-risk pregnant women receive perinatal services. The Kaunas Regional Ethics Committee approved the study, protocol No. BEC-LSMU(R-14). The possible cases of uterine rupture were identified using double data sources—first, from medical database records and second, manual identification about each case of uterine rupture from medical records. 

Inclusion criteria: in this article, uterine rupture is defined as a case in which an emergency CS had been performed due to specific clinical symptoms, and uterine rupture was confirmed during laparotomy. Both complete (*n* = 22) and partial (*n* = 13) uterine ruptures, which are also often referred to in the literature as uterine scar rupture, were analyzed. Complete uterine rupture is a tearing of all uterine layers. Partial rupture was considered when the serosa layer is intact.

Exclusion criteria: Asymptomatic cases (*n* = 3) in which a uterine muscle defect was found during elective CS were excluded from the analysis as cases that occurred until 22 weeks of gestation.

In this study, we examined risk factors for complete and partial uterine rupture after starting labour, and clinical symptoms of suspected complete or partial uterine rupture were analyzed separately. Risk factors for uterine rupture (complete and partial) for those with a previous cesarean delivery (*n* = 5626) were analyzed separately, too. The control group is represented by all births delivered in our department from 2004–2019 (*n* = 51,525). Statistical analysis of the survey data was performed using IBM SPSS Statistics 24.0 software. Descriptive statistics—absolute (*n*) and percentage frequency (%) distribution were used to evaluate the distribution of the analyzed symptoms in the selected sample. Quantitative data are presented as arithmetic mean ± standard deviation (SD). The relative risk and absolute risk were calculated using risk analysis. Spearman’s correlation coefficient was calculated for the relationship between the number of births and the frequency of uterine ruptures. A value of *p* < 0.05 was considered significant.

## 3. Results

During the 15 years 51,525 women gave birth at *Kauno klinikos*, 35 (0.067%) of them had a symptomatic uterine rupture. The total incidence of uterine ruptures is 6.8 out of 10,000 births or 1 in 1472 births. The mean gestational age with uterine rupture was 37.1 ± 4.6 (range 25–42) weeks of pregnancy, with a mean neonatal weight of 3095 ± 999 g. The most common clinical symptoms of suspected uterine rupture are shown in Table 1. Nine cases (26%) led to an emergency CS, when uterine rupture was suspected occurred in the second stage of labour.

In our study women delivered with a uterine scar after previous CS, laparoscopic myomectomy, and without any scar in the uterus (Table 2). Exceptional cases of uterine ruptures with intact uterus occurred in 2007 and 2008: the rupture of the bicornuate uterus at 25 weeks of gestation and prostaglandin-induced delivery at 26 weeks of gestation due to severe preeclampsia.

All cases and factors associated with uterine rupture are presented in Table 3. The risk of uterine rupture in a group after a previous CS was statistically significantly higher.

The study group had other (potentially) pregnancy complications and diagnoses also, which are presented in Table 4. 6 of 35 births (17%) resulted in perinatal fetal death (one of which was diagnosed before the onset of labour due to placental abruption, three of them during the labour with a uterine scar after one CS, one of them had bicornuate uterus at 25 weeks of gestation and one underwent induction of labour at 26 weeks of gestation due to severe preeclampsia). Neonatal mortality reached 22% (*n* = 5) among the complete ruptures and 7% (*n* = 1) among partial ruptures. The number of cases of severe hypoxia (Apgar <6 points after 5 min) was similar: 18% (*n* = 4), among the complete ruptures and 7% (*n* = 1), among partial ruptures. One patient underwent hysterectomy (uterine rupture occurred in a case with placenta previa and uterine scarring after CS). Maternal deaths were prevented. The mean blood loss was 1415 ± 1096 mL (range 400–5300 mL), and transfusion of blood components was required in four patients (11%).

Uterine rupture cases that occurred during labour after a single previous CS were analyzed separately. During the study period, 5626 women who had a scar in the uterus after one CS gave birth in our department, 2126 (38%) of them gave birth by natural pathways. The 25 analyzed cases of the symptomatic uterine rupture (after one CS) increase frequency during the study period to 44.4 out of 10,000 births, the estimated absolute risk is 1 in 225 births. Four patients gave birth less than 24 months after previous cesarean delivery. The trends of deliveries after one CS and uterine rupture over the years is shown in Figure 1. It shows a medium positive correlation between the number of births with a scarred uterus and the frequency of uterine ruptures *r* (16) = 0.63, *p* = 0.009. The extreme increase in the number of uterine ruptures in 2019 is possibly due to better data availability through a local electronic medical database and the cumulative effect of other possible risk factors.

The mean gestational age at which the uterine rupture after one CS occurred was 38.8 ± 3.4 (range 28–42) weeks, with a mean neonatal weight (in single pregnancies) amounting to 3325 ± 640 g. It has been estimated that if there is a uterine scar after one CS, the chances of uterine rupture are significantly increased upon augmentation of labour and regional analgesia (Table 5).

## 4. Discussion

The incidence of symptomatic uterine ruptures at our tertiary referral centre is similar to other developed countries, where the incidence of uterine ruptures ranges from 1.9 to 38 cases per 10,000 births. Maternal mortality due to uterine rupture is about 0–1.4%, and stillbirths at about 12% [1,2]. The incidence of perinatal mortality due to uterine ruptures in hospitals, where less than 3000 women give birth per year, is statistically significantly higher than at tertiary centres [4].

Based on the latest literature source, the most consistent early indicator of uterine rupture is changes in CTG (prolonged, persistent fetal bradycardia) [6]. The most commonly identified symptom in this study was acute abdominal pain, which was reported by approximately half of the cases. Uterus scarring after previous a CS (especially in women who did not give birth by natural pathways) and induction of labour using prostaglandins are the most important risk factors for uterine rupture. According to the literature, the likelihood of uterine rupture is also increased by the short period (<12 or <24 months) after previous CS, augmentation of labour using oxytocin, abnormal fetal position, an excessive amount of amniotic fluid, abnormally invasive placenta (especially placenta increta and placenta percreta), placental abruption, connective tissue diseases, adenomyosis, trauma, uterine abnormalities [1,3,4,7,8] and even decreased myometrial scar thickness on ultrasound (less than 2.8 mm) [9]. However, there are other large-sample studies which indicate that births by natural pathways increase the risk of symptomatic uterine scar rupture after one CS by only 0.27%. Previous studies indicate that 370 electice CS would need to be performed to prevent only one symptomatic uterine rupture [2]. More frequent uterine ruptures have also been associated with hysteroscopy, in which uterine perforation has been performed, or the uterine septum has been removed as well as laparoscopic myomectomy [10,11,12]. Analysis of the association between high neonatal weight and uterine rupture showed controversial results. Different large-sample studies show opposite results due to a statistically significant effect on the higher incidence of uterine rupture when the neonate weight is more than 3500 g [1,13].

Most of the above-mentioned conditions are also diagnosed in the study group. The results of the statistical analysis also support the claim that labour augmentation in the presence of scarred uterus increases the chances of uterine rupture, but it is difficult to explain why regional anaesthesia has a similar effect [14]. It is estimated, that after epiduralanalgesia, patients feel less discomfort during labour augmentation; therefore, it is applicatory more often, higher doses of oxytocin may be more boldly administered and augmentation of labour continues for longer [15].

The significance of some risk factors is questionable due to the small sample size of the study. For example, induction of labour using prostaglandin (Misoprostol) in the presence of uterine scar after previous cesarean delivery. According to the literature, this increases the incidence of uterine ruptures, about 2.45% [16]. In 15 years, only one case of uterine rupture after administering prostaglandins in the presence of uterine scar after CS was recorded at our department (in 2004, induction of labour at 42 weeks of gestation) because this method of labour induction has recently been used only in exceptional cases (a total of 63 cases over 15 years). The bias of this study is the retrospective design, and there was missed or law quality information in medical records without universal standardized registration of cases during the study period. For these reasons, it was not possible to assess the effect of patient’s age, body mass index (BMI), socioeconomic factors, or smoking and length of delivery as risk factors for uterine rupture. On the other hand, the results of most studies support the elementary sequence that uterine ruptures are more common in patients older than 35 years who gave more births and underwent more CSs [1].

The main concern of this study was the discrepancy between the surgery findings and the formulation of the uterine rupture diagnosis. For the sake of objectivity, a distinction must be made between what we call a true uterine rupture—a complete separation of uterine muscle integrity where the amniotic fluid, amniotic sac, placenta or parts of the fetal body are found in the peritoneal cavity—and cases where the site of uterine muscle separation is restricted to a particular connective tissue (e.g., the covering bladder, a formed hematoma, or simply the remaining serous lining of the uterus). The importance of the size of the defect is also worth considering. The technique and the type of uterine suture is also an important risk factor of uterine rupture [13]. According to literature source, a single-layer uterine suture should be avoided for patients who could contemplate future vaginal delivery after cesarean section as it carries more than twice the risk of uterine rupture than double-layer closure [13]. Over a 15 year period, the uterine suture and the material could be changed, but we missed such information in medical records, so we could not evaluate this potential risk factor. Due to these discrepancies, the data should be evaluated critically. For the accuracy of the data, it would be useful to continue the analysis of uterine ruptures with larger-scale long-term multi-centre studies.

## 5. Conclusions

The incidence of uterine rupture at the tertiary referral centre is 6.8 cases per 10,000 births. In mothers with a scarred uterus after a single CS, the incidence of uterine ruptures is much higher, exceeding 44 cases per 10,000 births. The most significant risk factors for uterine rupture include uterine scarring and augmentation of labour or epidural anaesthesia in a previous cesarean delivery, also the most commonly reported clinical symptom was acute abdominal pain during delivery.

## Figures and Tables

**Figure 1 medicina-56-00574-f001:**
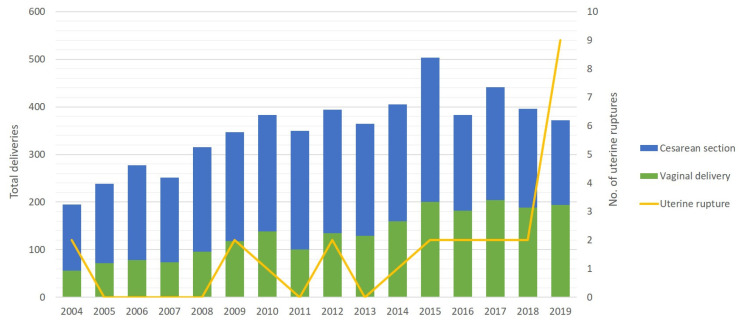
Distribution of women with a uterine scar after one CS per year.

**Table 1 medicina-56-00574-t001:** Clinical symptoms of suspected uterine rupture.

Symptoms	Incidence, *n* (%)		
	Total (*n* = 35)	Complete Rupture (*n* = 22)	Partial Rupture (*n* = 13)
Acute abdominal pain	21 (60)	15 (68)	6 (46)
Abnormal CTG	12 (34)	9 (40)	3 (23)
Vaginal bleeding	7 (20)	4 (18)	3 (23)
Unstable hemodynamics (hypotension, impaired consciousness, tachycardia)	2 (6)	2 (9)	0
Acute absence of contractions	1 (3)	1 (5)	0
Dystocia (failure to progress in labor or cephalopelvic disproportion)	4 (11)	2 (9)	2 (15)

CTG—cardiotocography.

**Table 2 medicina-56-00574-t002:** Distribution of the study group by scars in the uterus. CS: cesarean section.

Study Group		Incidence, *n* (%)
Without Uterine Scar		2 (6)
Scar in the uterus 29 (83)	Previous 1 CS	25 (71)
	Previous 2 CSs	3 (9)
	Previous 3 CSs	0
	Previous 4 CSs	1 (3)
Previous laparoscopic myomectomy		4 (11)

**Table 3 medicina-56-00574-t003:** Cases and risk factors of uterine rupture.

	Uterine Rupture			All Births	Relative Risk (95% Cl)			Absolute Risk
	Total	Complete Ruptures	Partial Ruptures		Total	Complete Ruptures	Partial Ruptures	
Spontaneous onset of delivery	31	18	13	43,345	1.1 (0.9–1.2)	1.0 (0.8–1.9)	1.2 (1.2–1.2)*	1 in 1398
Induction of labour:	4	4	0	8180	0.7 (0.3–1.8)	1.1 (0.5–2.8)		1 in 2045
Prostaglandin	2	2	0	3057	1.0	1.5		1 in 1529
Amniotomy	1	1	0	4503	0.3	0.5		1 in 4503
Oxytocin	1	1	0	462	3.2	5.0		1 in 462
Augmentation of labour	10	6	4	11,249	1.3 (0.8–2.2)	1.2 (0.6–2.5)	1.4 (0.6–3.2)	1 in 1125
Epidural analgesia	12	8	4	11,773	1.5 (0.9–2.4)	1.6 (0.9–2.8)	1.3 (0.6–3.0)	1 in 981
Preterm birth (<37 weeks)	9	9	0	7541	1.8 (1.0–3.1)	2.8 (1.7–4.6)*		1 in 838
Delivery after 42 weeks	1	1	0	98	15.0	23.9		1 in 98
Nulliparous	5	5	0	24,684	0.3 (0.1–0.7)*	0.5 (0.2−1.0)		1 in 4937
Multiparous	30	17	13	26,841	1.6 (1.4–1.9)*	1.5 (1.2–1.9)*	1.9 (1.9–1.9)*	1 in 895
Multiple pregnancy	2	2	0	1538	1.9	3.0		1 in 769
Uterine scar after CS	29	16	13	6852	6.2 (5.4–7.3)*	5.5 (4.2–7.1)*	7.5 (7.4–7.7)*	1 in 236
Total	35	22	13	51,525				1 in 1472

* *p* < 0.05, result is statistically significant.

**Table 4 medicina-56-00574-t004:** Incidence of pregnancy complications among all uterine ruptures.

Pregnancy Complications	Incidence, *n* (%)
Occiput posterior position	6 (17)
Abnormal fetal position (breech, transverse position)	5 (14)
Polyhydramnios	1 (3)
Obesity	3 (9)
Type 1 diabetes	1 (3)
Placenta previa	2 (6)
Placental abruption	1 (3)
Bicornuate uterus	1 (3)

**Table 5 medicina-56-00574-t005:** Possible risk factors for uterine rupture in the presence of a scarred uterus after one CS.

	Uterine Rupture			All Births at *Kaunas Klinikos*	Relative Risk (95 % Cl)			Absolute Risk
	Total	Complete Ruptures	Partial Ruptures		Total	Complete Ruptures	Partial Ruptures	
Uterine scar after one CS	25	14	11	5626				1 in 225
Induced labor	3	3	0	726	0.93 (0.3–2.7)	1.7 (0.6–4.5)		1 in 242
Prostaglandins	1	1	0	63	3.6	6.4		1 in 63
Amniotomy	1	1	0	351	0.6	1.1		1 in 351
Oxytocin	1	1	0	312	0.7	1.3		1 in 312
Augmentation of labor	10	6	4	857	2.6 (1.6–4.3)*	2.8 (1.5–5.2)*	2.0 (0.9–4.6)	1 in 86
Epidural analgesia	11	7	4	1032	2.4 (1.5–3.7)*	2.7 (1.6–4.6)*	1.7 (0.7–3.8)	1 in 94

* *p* < 0.05, result is statistically significant.

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
