# Peer review of "Symptomatic Uterine Rupture: A Fifteen Year Review"

_medicina, 2020, doi:10.3390/medicina56110574_

Round 1

Reviewer 1 Report

I found the manuscript really interesting and suitable to be published. I would like to suggest some issues to be fixed and some improvements to improve the manuscript.

Major points to be fixed:
A) materials and methods: "A value of P < 0.05 was considered significant [14]." Here ref 14 makes no sense

B) I noted that you had 0 uterine ruptures in the previous 3 CSs and 1 rupture in the previous 4 CSs it would be interesting to know how many previous 3 CSs and previous 4 CSs had a trial of labor in your population. I mean if you could in Table 3 stratify the "Uterine scar after CS" row according to the number of the previous CSs letting as to know the risk of scar rupture in previous 1, 2, 3, and 4 CSc separately in your population.

C) Page 2 line 83: "The mean gestational age with uterine rupture was 37.1 ± 4.6 (range 25 - 42) weeks of pregnancy, with a mean neonatal weight of 3095 ± 999 g."
Page 4 lines 121-122: "The mean gestational age at which the uterine rupture occurred was 38.8 ± 3.4 (range 28 - 42) weeks, with a mean neonatal weight (in single pregnancies) amounting to 3325 ± 640g." These two sentences seem to be contradictory please amend.

Author Response

Dear Reviewer,

We are grateful for your consideration of this manuscript.

We very much appreciate the reviewer for careful reading of our text and suggestions, which are have been very helpful in improving the manuscript. All comments we received in this study have been taken into account in improving the quality of the article, and we present our reply to each of them separately.

Reviewer 2 Report

This study is a valuable report on the frequency of uterine rupture and risk factors in a large number of 50,000 deliveries at a single institution.

Although the risk factors for cases of uterine rupture have been studied in detail, it is difficult to understand the specifics compared to previous reports, and it is necessary to examine each case one by one. In addition, although 1 CS has been considered individually, there are no data to show why 1 CS is worthy of consideration when compared to cases with more than 1 CS. It should be clarified whether it is because of the large number of cases or whether it is because the most recent CS is a likely risk factor for uterine rupture. If the data collection is now accurate and the number of cases of uterine rupture is increasing in 2019, then the risk of uterine rupture is likely to be quite high at 250 cases per 1000 people. Is there any reason for this?

Perinatal complications are mentioned in the results, but I didn't know what kind of assessment it is.

Since this is a retrospective report on a long period of time (15 years), it is undeniable that the criteria for induction a labor have changed over time, and I do not think it is necessary to discuss this in depth. We believe that the same is true for the surgical findings. We hope that this issue will be discussed in more detail in the future.

Table 3.

>The risk of uterine rupture in a group of post-term (>42 weeks) births and births after a previous CS was statistically significantly higher.

The results appear to differ from the results marked with an asterisk. You need to be sure that the statistical analysis is accurate.

Table 4 is not considered unnecessary as it does not appear to be information that would affect the results or the discussion.

I look forward to a constructive exchange of ideas.

Author Response

Dear Reviewer,

We are grateful for your consideration of this manuscript.

We appreciate the reviewer very much for the careful reading of our text and suggestions, which are have been very helpful in improving the manuscript. All comments we received in this study have been taken into account in improving the quality of the article, and we present our reply to each of them separately.

Round 2

Reviewer 2 Report

>Response 1: We believe, that extreme increase in the number of uterine ruptures in 2019 is possibly not only due to better data availability through a local electronic medical database, but due to the cumulative effect of other risk factors too. For example, after epidural anesthesia, which is used more often, patients feel less discomfort during labor augmentation, therefore it is applicatory more often, higher doses of oxytocin may be more boldly administered and augmentation of labor continues for longer. Frequency of uterine ruptures of scarred uterus would be around 237 cases per 10000 births in 2019, compared to 44.4 out of 10000 births during the whole study period.

I think you're right about the reasons for identifying the 1CS separately.

Are you saying that the significant increase in the risk of uterine rupture in 2019 over the general risk of morbidity is because of the increase in epidural anesthesia? Obviously, I think it's more than previous report so far, but does that mean that this is the current situation?

>Response 2: Perinatal complications include all the negative outcomes discussed in the article related to the patient and the fetus (for an example, perinatal fetal death - line 100, severe hypoxia - line 104, hysterectomy - line 106, blood loss - line 107 and other complications of pregnancy, provided in the Table 4).

What can you say differently about that result compared to the previous report? Is it the same as what we have been reporting?

>Point 3: Since this is a retrospective report on a long period of time (15 years), it is undeniable that the criteria for induction a labor have changed over time, and I do not think it is necessary to discuss this in depth. We believe that the same is true for the surgical findings.

>Response 3: The most important change, associated with labor induction, over the period of 15 years, is regarding the frequency of trials of labor with a scarred uterus after 1 previous CS. It affects the frequency of uterine ruptures undeniably. A medium positive correlation between the number of births with a scarred uterus and the frequency of uterine ruptures r (16) = 0.63, P = 0.009 is calculated. Definitely further trials and discussions are desirable.

I understood the link between the number of deliveries and uterine rupture, but my point was about the discrepancy between the surgical findings and the formulation. Is there a statement about that in the results? Are there any reports that support the importance of the surgical findings, the size of the defect, and the importance of the suture technique?

>137-138

According to the literature, the most common clinical symptom warning of an ongoing uterine rupture is pathological CTG (usually fetal bradycardia) [6].

I'm sorry I missed that. The paper reported that the most consistent early indicator of uterine rupture is the onset of a prolonged, persistent, and profound fetal bradycardia. They are not discussing it as a general clinical findings. If you wish to discuss the frequency of clinical symptoms, it is advisable to cite appropriate review articles.

Author Response

Thank you for your revision,

All comments we received in this study have been taken into account in improving the quality of the article, and we present our reply to each of them separately.

Hope you will be satisfied with the answers.
